# Strengthening University Student Wellbeing: Language and Perceptions of Chinese International Students

**DOI:** 10.3390/ijerph17155538

**Published:** 2020-07-31

**Authors:** Lanxi Huang, Margaret L. Kern, Lindsay G. Oades

**Affiliations:** Melbourne Graduate School of Education, University of Melbourne, Melbourne 3010, Australia; peggy.kern@unimelb.edu.au (M.L.K.); lindsay.oades@unimelb.edu.au (L.G.O.)

**Keywords:** wellbeing, lay perspectives, language, Chinese international students, tertiary education, mental health, wellbeing literacy

## Abstract

Students at the tertiary education level in Australia are at increased risk of experiencing high levels of psychological distress, with international students at particularly high risk for poor adjustment. As mental health and wellbeing strongly correlate with students’ academic performance and general overseas experience, a growing number of studies focus on what universities can do to effectively support students’ wellbeing. However, assumptions are made about what wellbeing is, strategies primarily focus on treating mental ill-health, and treatment approaches fail to account for cultural differences. This study aimed to explore how Chinese international students understand wellbeing, the language used about and for wellbeing, and activities that students believe strengthen their own and others’ wellbeing. Eighty-four Chinese international students completed the online survey, and a subset of 30 students participated in semi-structured interviews. Data were analysed using thematic, phenomenographic, and language analyses. Physical health and mental health appeared as the key components that participants believed defined wellbeing, and intrapersonal activities were perceived as the primary approach used to strengthen wellbeing. Findings help broaden the understanding of wellbeing concept from the population of tertiary students, identify students’ perspectives of activities that strengthen their wellbeing, offer a snapshot of the language used by Chinese students around wellbeing, and provide new data of population health through a wellbeing lens.

## 1. Introduction

The coronavirus pandemic of 2019 and 2020 has brought public health into the spotlight, emphasising the extent to which physical health matters. International travel stopped, economies entered deep recessions, millions of workers lost their jobs, and the social fabric of society came to a standstill, all centred around flattening the infection curve and preventing the possibility of overwhelming health care systems and the unnecessary loss of human lives. However, even as the infection curve flattened, awareness began to shift to a second curve: mental illness. Mental health/illness refers to people’s cognitive, emotional and behavioural status, which affects how people think, feel, behave, and interact with others [1]. Wellbeing goes beyond the absence of mental illness, and is constituted by comprehensive physical, mental, emotional, and social health factors relating to feeling good and functioning well [2]. Mental health and wellbeing are critical concerns. Although public health and medicine have excelled at preserving life, the quality of that life must also be considered [3].

Even before the pandemic began, the quality of many people’s lives has been questionable. Within any given year, an estimated one out of every five individuals will experience severe mental illness, with some populations at higher risk than others [4]. One of these at-risk populations are young adult students. Among all of the populations that suffer from mental disorders, students at the tertiary education level have a higher incidence of mental health problems -- including anxiety, depression, and other mental health conditions - than other age groups [5]. For instance, within Australia, a survey of over 3,000 students indicated that more than half reported high or very high levels of psychological distress, and one-third had thoughts of self-harm or suicide [6]. Many students experience academic stress and related performance expectations [7]; lack of social support networks, unfulfilled desires for friendship, and sexual harassment/assault [8]; worries over financial situation and imbalances between work and study; and higher likelihood of unhealthy lifestyles, including poor diet, alcohol use, and insufficient sleep [6]. All of these increase the risk of psychological distress and mental illness. 

International students are at even greater risk than local tertiary students, as they live away from home, navigate the many challenges of living and studying in a foreign culture, and try to balance the many demands encountered, often with limited support. International students represent a sizeable population; for instance, there were over 600,000 international students studying in Australia in early 2020, with 28% of these students from China [9]. Despite greater risk for mental illness, international students are less likely to seek professional help than domestic students, with language barriers and cultural-based stigma identified as leading causes of not seeking help [10]. Models of care are primarily rooted in Western methodologies, which can be insensitive to cultural differences. Simply improving existing mental health services is insufficient; greater understanding of how students think about and talk about mental health and wellbeing is needed to better meet the needs of this important population. 

According to Oades and Johnston [11], wellbeing literacy refers to how people control the use of language around wellbeing. This includes the language that people use about wellbeing and how language is used to build and support wellbeing. That is, there is a need to focus on understanding and developing the wellbeing literacy of international students. This study focuses on the first need: understanding wellbeing literacy. The current study examines mental health and wellbeing from the perspective of Chinese international students in Australia, identifying the role that language plays in students’ understanding and experience of wellbeing. 

### 1.1. The Mental Health Burden

Mental illness has significant impacts on populations worldwide. As of 2017, an estimated 792 million people suffered from mental or neurological disorders [12], with at least one in four people affected by mental illness at some point in their lives [13]. At least 1/3 of disabilities are due to mental illness [14], with incidences of mental illness most likely underreported. Mental illness not only affects an individual’s functioning and quality of life, but also affects families and communities [15,16]. Globally, an estimated $1 trillion is lost in productivity every year due to mental illness, with the economic impact of mental illness expected to exceed $2.5 trillion by 2030 [17].

In Australia, the Australian Bureau of Statistics (ABS) found that 4.8 million Australians suffered from a mental health disorder in 2017, with around 45% of Australians having experienced a mental disorder in their lifetime. Over 3000 individuals engaged in self-harm behaviours, and 94.2% of respondents who reported suicidal ideation or attempts has experienced mental disorder within the previous 12 months. Mental illness had an estimated economic impact of $4,000 per person, costing the Australian economy $60 billion in 2016 alone [18]. 

### 1.2. Barriers to Care

Importantly, despite the mental health burden, many individuals do not receive proper professional care. Although treatment rates increased steadily from 37% in 2006 to 46% in 2016, over half of the population did not receive treatment. Rates are disproportional; one study found that young people were less likely to seek professional help, and 31% of women versus 13% of men received support and care [19]. 

Despite large amounts of human resources and financial support invested in this area, there are numerous obstacles that prevent people from receiving care. First, there are challenges in fully identifying those who suffer, with many people being hesitant to report concerns. Mental illness is often first detected through self-reported screening instruments, and yet such measures are prone to people adjusting answers to socially desirable responses, lack of self-insight by respondents, distorted interpretation of the questions, and sampling biases that exclude those who are most at risk for disorder [20,21,22]. Second, many people do not seek treatment, impacted in part by stigma and discrimination [23,24]. Third, for the people who do seek care, available services often inadequately meet the needs that people bring, with shortages of health care human resource, unclear efficacy of various treatments for different populations/communities, and barriers towards providing equitable care for individuals with economic and social vulnerabilities [25,26,27].

To address these barriers, numerous policies and practices have been trialled. A primary approach involves public education and awareness campaigns to reduce barriers of stigma and discrimination [13]. Other efforts have focused on developing the capacity of the mental health care systems, such as incorporating KidsMatter and MindMatters programs within Australian schools to proactively support mental health [28]. Taking advantage of online technologies, a growing number of programs offer self-help interventions [29]. For such approaches to be effective, it is necessary to ensure that resources and approaches are sensitive to the needs of different subpopulations, rather than assuming that one size will fit all.

### 1.3. International Students as an Important Population 

Amongst the many populations that could be considered, only limited focus has been given to international students. Given the recent decades of internationalisation and globalisation, a large number of students choose to study and live abroad, with study periods ranging from a few weeks overseas to completing multiple degrees abroad. In 2019, there were more than 950,000 international students who enrolled in higher education, Vocational Education and Training (VET), schools, English Language Intensive Courses for Overseas Students (ELICOS), and non-award sectors in Australia [9]. At the beginning of 2020, Australia was the third most popular education destination, following the United States and the United Kingdom, and hosts students from 204 countries [30]. Compared to other countries, Australia has the largest percentage of international students amongst the total higher education population, with 28% of students being international [31]. In the international education market, approximately 53% of total international students come from central Asia, with China being the largest source [32], accounting for around 28% of total international student population in Australia in 2019 [9]. Currently, international education is the fourth largest export, contributing more than AUD37 billion and providing 250,000 jobs opportunities across the Australian economy [33]. Besides being a valuable financial asset, international students also promote cultural prosperity, bring diverse heritage and perspectives, enhance cultural awareness and appreciation, contribute to the intellectual capital, and provide a workforce for the host country. 

For international students, while studying abroad provides an education degree and overseas experience, it also often brings many challenges and difficulties. Compared to domestic students, international students have been identified as a high-risk group for mental health and wellbeing issues, with more numerous/severe transition problems [34]. Students face academic challenges, language barriers, financial difficulties, social isolation or lack of social support, cultural adjustment, perceived discrimination or prejudice, homesickness, and other practical issues associated with changing environments [35,36,37]. These additional challenges, often without support by others and the coping strategies that might have been useful in the home country, increases risk for a number of mental health issues [38]. For example, one study found that of Chinese international students in a given university, 45% reported symptoms of depression and 29% reported symptoms of anxiety [39]. The Victorian Coroners Prevention Unit confirmed at least 27 suicide deaths of international students in Australia between 2009–2015 [40]. Further, untreated mental illnesses are likely to have significant implications for academic success, substance use, and social relationship [41]. Consequently, the demand for mental health and wellbeing support has become a significant concern for the international student population.

Despite the widespread availability of counselling services, mental health problems are often unnoticed or overlooked among international students, such that students suffer in silence [42]. For instance, evidence indicates that only 17% of Chinese students accessed counselling service, compared to 33% of their Australian peers [43]. A number of factors create significant barriers. First, studies find that international students often experience difficulty in understanding mental health and wellbeing information, struggle to recognise and express the symptoms of mental disorders, and lack ability to understand and express their feelings in the non-native language setting [44,45,46]. This speaks, in part, to the literacy of international students about their wellbeing. High levels of health literacy, mental health literacy, and emotional literacy can increase access to the proper professional care and the treatment efficacy, and help people to enjoy better physical and mental health outcomes [47,48,49]. Yet it is questionable the extent to which the wellbeing literacy of international students matches the wellbeing literacy assumed by healthcare systems.

Second, stigma remains a major barrier [50]. Stigma towards mental illness can be seen in almost every culture of the world. Mental health issues like depression and anxiety have been widely viewed as personal weakness instead of illness. People with mental health issues may also be labelled as “crazy” or “an idiot”, which influence their attempt to seek professional help [51]. In the Chinese context, there is a cultural tendency toward emotional inhibition and beliefs of “no suffering no gain”, which increase the stigmatisation of mental health issues [52]. 

Third, international students are challenged by having to negotiate the local health care system, often while lacking awareness and knowledge about the details of health insurance and system procedures [53]. Fourth, mental health care provided in the university has been criticised as help-seeking and access to the care do not meet the needs of student populations, influenced by the barriers of financial constraints, attitudes and knowledge about services, and concerns about privacy [54]. Research also shows that individual needs and cultural barriers are often neglected during mental health counselling sessions [55]. 

### 1.4. A Need to Understand Students’ Perspectives

As described above, studying overseas provides numerous opportunities for young adults, but also brings a number of challenges. There are persistent gaps between the wellbeing issues faced by international students and the limitations of current mental health supports provided by the education institutions. Simply improving the existing mental health service is not enough to overcome the identified barriers. It is crucial for tertiary educational institutes to develop their awareness/understanding of mental health issues and wellbeing affecting international students, provide accessible wellbeing resources, reduce stigma, and provide professional mental health and wellbeing support.

An important question is how to improve international student’s wellbeing in an accessible and effective way. The lack of sufficient care is driven in part by misunderstanding and lack of knowledge around how these students think about and talk about mental health and wellbeing. If we can better understand international students’ language, then we might better be able to meet their needs and improve the mental health of this sizeable population. However, little is known about how university students’ wellbeing is experienced and strengthened, particularly from the students’ perspective. The current study aimed to: (1) explore the language used about and for wellbeing and (2) identify activities that students believe strengthen wellbeing.

## 2. Materials and Methods 

### 2.1. Design

Qualitative data were collected through on online survey, complemented by in-depth interviews with a subset of respondents. All procedures were approved by the University of Melbourne’s ethics review committee (#1954456.1). Data were collected between September 2019 and February 2020. Chinese International Students Associations and other similar associations/student groups from the eight universities in Melbourne were contacted via email and WeChat, with an invitation to share the study details with eligible students. 

Students who were interested in the research could go to the online survey, read details about the study, and decide to participate. The online survey began by collecting basic demographic information and screening for mental illness. As the study intended to focus on a normal (nonclinical) population, the survey included the six-item Kessler Distress Scale [56], which has been used in a number of population-based studies to indicate potential psychological distress [57,58,59]. The scale is meant to be used only as a screening instrument, not as a diagnostic tool. If a potential participant scored at 19 and above, they were excluded from the study but referred on for further evaluation and care.

The survey then proceeded with a number of quantitative and qualitative questions. Demographic variables included gender, age, English proficiency level (reading, writing, comprehension), length of stay, education level, and their education status. Gender, age, education level, and education status were assessed on categorial scales. English proficiency levels were assessed in text-entry format question, and English test scores were converted into IELTS scores, based on the equivalent standard set by the university’s admission requirements. 

Participants were then asked in an open-question format to describe the activities that they engaged in, are engaging in, and plan to engage in to maintain and promote wellbeing, for themselves and for others, and to identify the activities they do to experience wellbeing. 

At the end of the survey, participants were invited to indicate willingness to be involved in an interview to further explore their experiences. Willing participants were interviewed online using Zoom or WeChat, with interviews lasting 40 to 60 min. Participants could complete the questionnaire and interview in English, Mandarin, or a mix of English and Mandarin. Semi-structured interviews explored how students described and conceptualised wellbeing, and actions taken to support the wellbeing of self and others (e.g., “What does wellbeing mean to you?”; “What words can you think of to communicate/describe about wellbeing?”).

### 2.2. Participants

To be included in the study, participants had to be a Chinese international student who was (1) studying at a tertiary education institution in Melbourne, Australia, (2) at least 18 years old, and (3) had lived in Melbourne for a minimum of three months and a maximum of four years, to ensure that participants were beyond the immediate transition period and had significant experiences across the Australian and Chinese cultures. 

Of 233 individuals who showed interest in being a part of the study, 47 were excluded due to not meeting the inclusion criteria, and 32 were excluded due to showing signs of potential severe psychological distress. Of those who met the inclusion criteria, 84 students (27 males, 57 females) completed the online survey. On average, participants were between 21–29 years old. Of the 84 participants, 30 took part in semi-structured interviews. Table 1 summarises demographic characteristic details for the online survey and semi-structured interviews, respectively.

### 2.3. Data Analyses

Analyses focused on the free-response questions in the survey and interviews. For the survey data, qualitative content analysis was used, which involved (1) preparation: selecting the unit of analysis and making sense of the data; (2) organising: open coding, coding sheets, grouping, categorisation, and abstraction; and (3) reporting: summarising and sharing categories [60].

Interviews were transcribed and then analysed using a phenomenological approach, aimed at describing, analysing, and understanding Chinese international students’ experiences with and understandings of various wellbeing aspects of their worlds. The interview data are presented based on the phenomenological principle of “to the things themselves” [61], which were analysed following steps outlined by Dahlgren and Fallsberg [62] for phenomenographic data coding and analysis: (1) familiarisation, (2) condensation, (3) comparison, (4) grouping, (5) articulating, (6) labelling, and (7) contrasting. The oral data from the interviews were transcribed into a set of systematic written data, which were edited to be more readable and to convey meaningful information to the readers. The verbatim transcripts were edited to a polished, coherent style, which removed loose ends and ramblings. Data were classified into corresponding patterns, and then combined and catalogued into subthemes, with representative quotes of each subtheme presented here.

As most interviews (29/30) were conducted in Mandarin Chinese, analyses were conducted in Chinese, and then translated into English for writing and presenting. A translator with NAATI certification assisted with the translation process to minimise researcher bias and subjective interpretation. All the cited quotations have been verified by the translator, and the phrases with specific cultural meanings have been explained in detail within their particular context by providing the original expression in Chinese. 

## 3. Results

### 3.1. Revealing Language for and about Wellbeing

The first aim was to reveal language that Chinese international students use for and about wellbeing. In the interviews, participants were asked: “when you think of wellbeing, what do you think?” and “what does wellbeing mean to you?”. Analyses uncovered five main themes, which further could be classified into 22 subthemes. Table 2 summarises the themes, subthemes, provides representative quotes, and notes the number and percentage of times the subtheme was mentioned. 

Participants generally viewed wellbeing as being multidimensional in nature. Most commonly, participants pointed to physical and mental aspects, noting for instance that “*the first thought of wellbeing is related to physical and mental health*”, “*without physical and mental illness*”, and “*it is a positive physical and mental status*”. Nine participants mentioned happiness, noting for instance “*when I happy every day. if I am not happy, then I am not wellbeing*”. Forty percent of participants pointed to a sense of security, especially pointing to financial aspects, while others pointed to a sense of prosperity. One student expressed “*wellbeing to me means I can afford all the cost by myself, as I financially depend on my parents currently*”. Other students believe that the “*financial condition*” and “*a reasonable income*” is the key to living and being well. Relationships were also commonly mentioned, noting for instance, “*you have a very good support system to back you up, no matter it’s family, friends, or partner, you need to have it so that you won’t feel insecurity*” and another noting the importance of having “*a healthy social interaction*”.

In the interviews, participants were asked to reflect upon “what words can you think of to communicate or describe wellbeing?” Figure 1 visualises the 54 meaningful words and phrases that were mentioned, with larger words indicating greater frequency (min = 1, max = 22), and no specific meaning for the colours. Aligned with the themes described above, “happiness”, “physical health”, “mental health”, “love”, and “security” were the most frequently mentioned words/phrases.

Participants were then asked to “describe a person with high wellbeing level” and “describe a person with low wellbeing level”. Figure 2 illustrates the 57 words/phrases mentioned for high wellbeing (left) and the 74 words/phrases mentioned for low wellbeing (right). High wellbeing was seen in “hobbies” “passion”, “family”, and “positive”. Low wellbeing included a broader range of words, but numerous words pointed to negative emotions and cognitions, and poor physical and mental health.

To further explore the role that language plays in understandings of wellbeing, participants were asked to “translate wellbeing into Chinese”. As illustrated in Figure 3, nine translations were identified, pointing to the range of ways that students understand a single word. Students commonly expressed the difficulty of accurate translate “wellbeing” into Chinese. For instance, one student noted: *“It’s hard because I couldn’t find a suitable word to cover the meaning of wellbeing. I think there are so many things in it. It is difficult to describe and summarize wellbeing in a simple word.”*


Some participants focused on the components of wellbeing, such as “mental health + physical health”, “happiness + safe + health”, and “life satisfaction”. Others provided literal translations, suggesting that the “well” reflects “happy”, “good”, “positive”, or the “ideal”, and the “being” represents the “status” or “a long-term stage”.

The struggle to relate their Chinese translation to the word “wellbeing” reflects a blurred conceptualisation of the concept. Seven students noted that wellbeing had the same meaning to them in English and Chinese. Three highlighted that their knowledge of wellbeing came from English-speaking information, which shaped their understanding of the concept and further influenced their Chinese translation. Four participants expressed frustration with the question, two of them mentioned that the context should be taken into consideration, as the meaning of the word could be totally different. Another suggested that the word should only be used in English, as the Chinese translation is “*too weird in daily conversation*”.

As summarised in Figure 4, the interviews suggested that the wellbeing construct takes on very different meanings in English versus Chinese. In terms of scope, participants perceived wellbeing as broad and abstract in English, versus materialistic and specific in Chinese. Wellbeing was perceived as focussing on the self in English versus others in Chinese. For instance, one participant noted: “*I will think more about the interpersonal relationships, such as friends and family in Chinese. But if I think it in English, I will consider more about myself, like how to manage myself*”. Wellbeing was seen as neutral and fundamental in English, versus being positive and ideal in Chinese. For instance, one student noted “*when I think it in Chinese, I will think it to be more positive. But if I listen to the word in English, I will think it more neutral, which could possibly include both positive side and negative side*”. Finally, wellbeing was seen to be dynamic and long-term in English versus static and short-term in Chinese. 

### 3.2. Activities that Strengthen Wellbeing

In the online survey, participants were asked to reflect upon activities that they engage in to support their own wellbeing. Analyses uncovered six main themes, which further could be classified into 21 subthemes. Table 3 summarises the themes and subthemes, representative quotes, and the frequency that each theme was mentioned. At a broad level, 76% of the themes represented activities done alone (intrapersonal activities), while the remaining 24% involved drawing on different forms of social support (interpersonal activities). 

The main types of activities were cognitive or physical in nature. The most popular activity involved different forms of exercise. For instance, participants noted: “*I’m exercising every week to help maintain both mental and physical health. It can make me focus on the moment of sweating and let all the things go with sweat.*” Participants also pointed to the importance of continued learning and exploring, noting for example, “*I’d love to explore different suburbs in Melbourne on weekends*”, “*learn new skills*”, and “*experience new things*”. Interestingly, some participants pointed to studying and working hard to achieve goals as a pathway to wellbeing.

Participants also pointed to the benefit of seeking or providing support to others. Primary forms of support included friends, parents, and teachers. Participants noted that they would “*chat with my good friends about unhappy, stressful things to relieve inner anxiety*”, “*hang out with friends*”, “*spend more time with family*”, and “*video-chat with families at home*”. Notably, only 2% pointed to seeking professional help. 

In the survey, participants also reflected upon how they support other people’s wellbeing in the past, at present, and what they might do in the future. Four main themes emerged, which further could be categorised into 12 subthemes, as summarised in Table 4. Interestingly, while the majority of activities were other oriented, 18% of activities were self-oriented, including 2% who did nothing for others. 

“Speaking” was the most common subtheme, which included a number of different strategies. Twenty-two responses reflected offering care and comfort, such as “*ask them how their lives are going*”, while 11 students pointed to providing encouragement (e.g., “*encourage others to give them directions*”). Participants noted that they want to “*be someone who will answer them*”, “*ask them how their lives are going*”, “*encourage others to increase their wellbeing*” and “*deliver positive energy*”. Thirteen responses focused on providing perspective or advice, such as “*recommend good books*” or “*give some advices*”. Two responses pointed others to professional help (“*suggest them to see a counsellor*”), although there was a sense of doubt about whether it is appropriate or not. 

Responses also pointed to trying to be a good listener, noting “*being a listener*”, “*listen to other’s worries*”, and “*listening without interrupting*”. Some responses pointed to more indirect forms of support. For instance, when others need company, participants might “*ask people out for exercise*”, “*planned activities to engage the person who was low*”, and “*hanging out with them to create fun experiences*”. Participants also expressed their thoughtfulness: “*be kind, sensitive and empathy*”, “*help them while respect their autonomy*”, “*be sensitive to other’s emotion*”, and “*stand in other’s shoes*”.

## 4. Discussion

Mental health is increasingly becoming an important public health concern worldwide, with some populations simultaneously at risk for poorer functioning, but less likely to seek and receive proper care than other populations. International students are one of these high-risk populations. While this is caused by a number of factors, these risks may be compounded by the language used both in conveying mental health and wellbeing information and in offering and providing care. Cultural differences can impact upon how words are used and understood (e.g., [63,64]. That is, there may be mismatches between the wellbeing literacy assumed by our systems and the wellbeing literacy of international students living within that system. There is a need to better understand the language used about and for wellbeing. The current study addressed this in one population—Chinese international students living in Australia. 

The findings show that “mental health/illness” and “physical health/ illness” are crucial components and phrases when Chinese international students understand and talk about wellbeing. Participants had different understandings of the wellbeing concept in English and Chinese. Participants also revealed a range of activities that they use to support their own and others’ wellbeing. 

### 4.1. Language about and for Wellbeing

Chinese international students perceived wellbeing as a multidimensional concept, including mental health, physical health, security, social relationships, and prosperity as key components. This was further evident in the specific words and phrases used to describe wellbeing. Interestingly, high wellbeing was seen more by what a person does (e.g., hobbies, fitness, passion, career), whereas low wellbeing was seen more in what people feel (e.g., depression, anxiety, unloved, purposeless).

Although the knowledge of wellbeing has generally been built upon studies from westernised and industrialised nations, a growing number of studies have explored the impacts of culture on wellbeing [65,66,67]. Supporting these studies, the current findings point to both similarities and differences between dominant Western conceptualisations of wellbeing and the themes revealed here. The multidimensional nature aligns with a growing consensus that wellbeing is multidimensional in nature (e.g., [68,69,70,71]). However, while multiple models place an emphasis on positive emotion (e.g., [71,72,73]), positive emotions were rarely mentioned. Instead, participants pointed more towards physical health, a sense of security, and prosperity. 

The results support the influence of self-construal processes [74] and emotional experiences [75] that have an impact on how wellbeing is understood. While Western models emphasise the need to reduce stress and pressure (e.g., [76,77,78]), participants pointed to the value of negative experiences such as pressure, which can be seen as a positive motivator for improving themselves. One interviewee noted that anxiety or depression at a subclinical level is okay and acceptable as long as it does not affect daily functioning too much. This perspective aligns with Buddhism, which is more concerned with mental imbalance and human suffering, viewing suffering as the nature of human life from the very start and a crucial source for security and wellbeing [79]. In Taoism practice, individuals are encouraged to appreciate the imperfection in life and embrace negative emotions and “both sides” of their life to achieve peace of mind [80]. 

Notably, the word “wellbeing” does not directly exist in Chinese and thus is hard to translate, which was reflected by comments and frustrations expressed by participants. According to the English Chinese dictionary, wellbeing has been understood as a combination of happiness, peace, and physical and mental health. Yet participants expressed their confusion and frustration when they considered how to translate wellbeing into Chinese. Further, participants had different conceptions for English and Chinese contexts, with wellbeing in English being more abstract, focused on the self, neutral, and dynamic in nature, versus being more specific, focused on others, positive, and stable in nature in Chinese. Considering “wellbeing” is often used throughout the public health literature, this might suggest that international students may have a very different understanding of the word, which could impact upon cognitions and behaviours. Another example is the Chinese translation of wellbeing as “being good”. Students noted that there is a moral bottom line or standard and to be civilised to achieve the status of being good. This idea mirrors the Confucian value of Wu Chang, in which social norms with virtues regulate behaviours for collective welfare, such as humanity, righteousness, ritual, knowledge and integrity [81]. Instead of mundane happiness, another critical idea relating to ideal personal status is Jun Zi (Chinese translation; “perfect man”), which refers to a perfect person, a combination of saint, scholar, and gentleman who ideally serves as a moral model for others [82].

### 4.2. Activities that Support Wellbeing

Participants identified a variety of activities that they engage in to support their own wellbeing, as well as pointing to ways that they provide support for others. Students particularly focused on physical and cognitive-based activities. Notably, supporting wellbeing was primarily seen as an intrapersonal activity, rather than something done with others. Others were seen as providing social support or help when needed, rather than as central to caring for wellbeing. This extended into how they supported others’ wellbeing, with the most common activity being providing emotional (e.g., listening, caring, encouraging) forms of support. This suggests that Chinese international students believed it was their own responsibility to maintain and promote wellbeing, rather than being dependent on others. 

The activities noted support cultural-based stigmas. According to Haslam [83], four dimensions of lay understandings of mental illness have been identified: pathologising, moralising, medicalising, and psychologising. The pathologising dimension sees mental illness is abnormal and deviant. The moralising dimension emphasises individual responsibility for the illness. The medicalising dimension highlights that mental illness has a somatic basis. The psychologising dimension ascribes mental illness to psychological dysfunction. Studies suggest that Western cultures tend to endorse the medicalising and psychologising dimension, whereas the pathologising and moralising dimensions are more embedded in Eastern culture [83,84,85,86,87]. The findings in the study align with the moralising dimension. For example, more than three-fourths of the descriptions for promoting one’s own wellbeing focused on self-based activities rather than seeking outside support, and strategies for supporting others focused on their own actions, such as becoming a better self, being kind and sensitive, and not troubling others. 

### 4.3. Implications

This study reveals language that Chinese international students use to understand and talk about wellbeing, as well as strategies that they use to support wellbeing in themselves and others. Mental health and wellbeing are words that are commonly used across the public health literature, but the question becomes the extent to which those words are meaningful to different populations. If words are meaningless, then educational messages, support, services, and approaches to proactively support functioning may be a waste of resources, or only communicate with local populations, fuelling inequities in support and care and further contributing to risk for already vulnerable populations. The visualised language around wellbeing clarifies and informs what words might be useful for communicating about wellbeing with words that everyday people can understand and connect to this concept. The subtle differences in how the wellbeing concept is understood in English and Chinese point to the need for cultural sensitivity when promoting wellbeing in diverse populations. It requires educational institutions and the public health sector to endeavour to increase their cultural and contextual awareness when engaging the public with various translations of this concept.

Aligned with the World Health Organization (1948), conceptualisations of health and wellbeing included physical, mental, and social aspects. For Chinese international students, a sense of security and prosperity also mattered. Unfortunately, a sense of security has been undermined for many Chinese internationals, due to discrimination and stigmatisation that occurred through the COVID-19 pandemic [88]. This would suggest that it will be important for tertiary education institutions in Australia to identify ways to provide safe and respectful environments for Chinese international students in the future. 

The wellbeing pathways revealed in this study can inform tertiary institutions on what students actually may need to support their wellbeing, which might differ from the typical services that are offered. The services around supporting students’ five components of wellbeing should be taken into consideration, such as offering healthy diet and lifestyle information, promoting mental health and counselling service, providing opportunity for cross-cultural interaction and activities, enhance campus security and non-discrimination environment, and educating students about employability opportunities. Many foreign students struggle to interact with locals and might only develop relationships with other foreign students. In the interviews, participants mentioned the hope for universities to provide opportunities to socially interact with locals and noted interest in workshops that introduce aspects of the local culture, such as slang, local foods, and festivals. They also mentioned that universities could provide more cross-culture activities to enhance the awareness of cultural diversity on campus, as some locals do not want to interact with international students when they been approached. Notably, participants focused more on intrapersonal activities rather than seeking support from others. This suggests that educating students with self-motivated and self-learning skills towards wellbeing is vital, such as highlights the importance of wellbeing during orientation week, offer workshops, training, and compulsory course around wellbeing. 

The pathways described here might provide a helpful starting point for supporting Chinese International student wellbeing, but most likely are insufficient, and care should be taken in generalising the findings. The data in this study were collected in the latter part of 2019 and early in 2020, before the COVID-19 pandemic impacted upon Australia. As such, the findings point to general experiences by international students, which may shift as the broader socio-political landscape of universities evolve. The pandemic has brought additional challenges to students living and studying abroad. A recent report pointed to problems with how information is disseminated, housing, needs for social inclusion, work, safety, and health [89]. These are important issues to consider, and the current study provides greater insights into students’ experiences studying abroad. Further studies exploring students’ perspectives and needs beyond the COVID-19 pandemic will be useful. 

## 5. Conclusions

Wellbeing is a crucial part of public health. Students’ language about and for wellbeing and their ideas on strategies that support wellbeing inform the messages and strategies needed to support Chinese international students well. Findings help broaden the understanding of wellbeing concept from the population of tertiary students, identify students’ perspectives of activities that strengthen their wellbeing, offer a snapshot of the language used by Chinese students around wellbeing, and provide new data of population health through a wellbeing lens.

## Figures and Tables

**Figure 1 ijerph-17-05538-f001:**
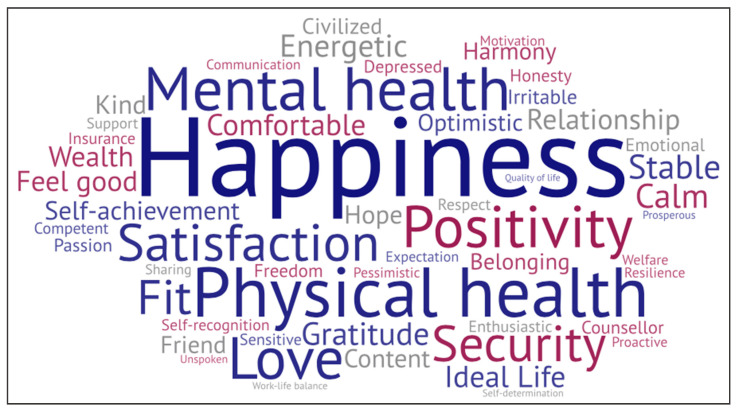
Words and phrases used by Chinese international students to communicate/describe wellbeing.

**Figure 2 ijerph-17-05538-f002:**
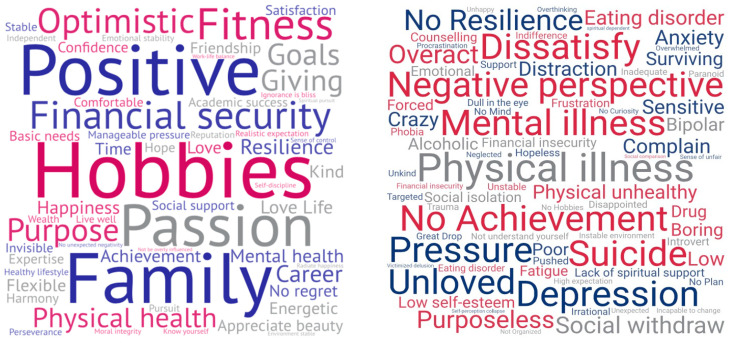
Words and phrases that participants used to describe a person high (**left**) and low (**right**) on wellbeing.

**Figure 3 ijerph-17-05538-f003:**
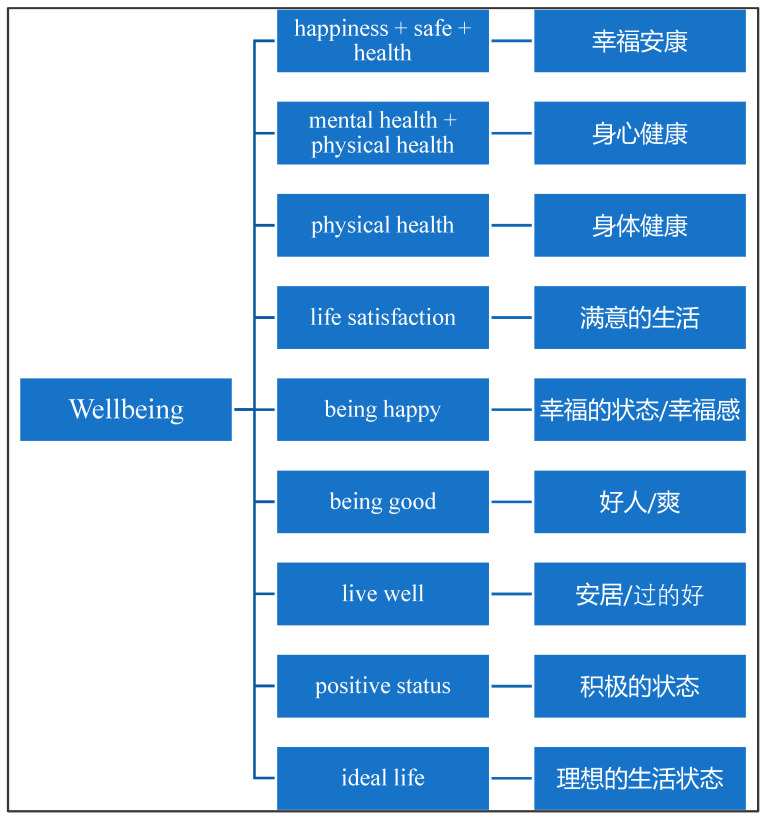
Participants’ Chinese translations of the word “wellbeing”.

**Figure 4 ijerph-17-05538-f004:**
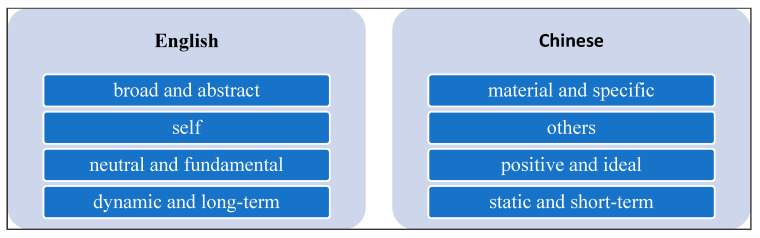
Chinese international students’ different understanding of wellbeing in English and Chinese.

**Table 1 ijerph-17-05538-t001:** Participants’ demographic characteristic details.

Characteristic	Category		Survey N (%)	Interview (%)
Gender	Male		27 (32.1%)	11 (36.7%)
	Female		57 (67.9%)	19 (63.3%)
Age	18–20		16 (19%)	1 (3.3%)
	21–29		60 (71.4%)	24 (80.0%)
	30–39		8 (9.5%)	3 (10.0%)
Education level	High school graduate, diploma/equivalent		20 (23.8%)	4 (13.3%)
	Undergraduate degree		20 (23.8%)	14 (46.7%)
	Graduate degree		44 (52.4%)	10 (33.3%)
Education Status	Foundation course/university credit		11 (13.1%)	1 (3.3%)
	Bachelor’s degree		22 (26.2%)	5 (16.7%)
	Graduate certificate or diploma		5 (6.0%)	1 (3.3%)
	Master’s degree		27 (32.1%)	15 (50.0%)
	Doctorate degree		19 (22.6%)	7 (23.3%)
English level	Comprehensive	Excellent	10 (11.9%)	9 (30.0%)
		Good	27 (32.1%)	12 (40.0%)
		Competent	29 (34.5%)	6 (20.0%)
		Modest	7 (8.3%)	1 (3.3%)
	Reading	Excellent	24 (28.6%)	14 (46.7%)
		Good	22 (26.2%)	7 (23.3%)
		Competent	19 (22.6%)	6 (20.0%)
		Modest	5 (6.0%)	1 (3.3%)
	Writing	Excellent	7 (8.3%)	4 (13.3%)
		Good	11 (13.1%)	8 (26.7%)
		Competent	37 (44.0%)	12 (40.0%)
		Modest	15 (17.9%)	4 (13.3%)
Length of stay	3–12 months		19 (22.6%)	2 (6.7%)
	13–24 months		21 (25.0%)	7 (23.3%)
	25–36 months		21 (25.0%)	10 (33.3%)
	37–48 months		23 (27.4%)	9 (30.0%)

**Table 2 ijerph-17-05538-t002:** Chinese international students’ understanding and language about wellbeing.

Main Themes	Subthemes	Representative Quotes	*n*	%
Mental based	mental health	“psychological sound, mental health is important”	20	67%
	happiness	“it’s happy and joy”, “it means I am happy everyday”	9	30%
	positivity	“I will have a positive attitude to face problem”	6	20%
	mental illness	“I will think about the opposite, like depression”	2	7%
	counselling	“I will think it’s a counselling workshop”	2	7%
	hope	“have hope in my heart”, “have hope in the future”	2	7%
	manageable pressure	“have appropriate pressure to improve myself”	2	7%
	appreciation	“appreciate beautiful things/people/scenery”	1	3%
Physical based	physical health	“physical health”, “no physical illness”	17	57%
	comfort	“a physical status that make people feel comfortable”	1	3%
Security	financial security	“have reasonable income”, “match basic needs”	7	23%
	stable environment	“adequate shelter” “stable living status”	2	7%
	liveability	“the comfortable level of living”, “living status”	2	7%
	health insurance	“including health cover, related pension insurance”	1	3%
Relationship support	social support	“relationship with classmates and teachers”	11	37%
	social integration	“can helping others and able to integrate into society	2	7%
Prosperity	life satisfaction	“satisfied with my life”	4	13%
	purpose in life	“have clear purpose in my life without doubts”	3	10%
	quality of life	“the quality of my life, the living standard”	2	7%
	passion	“find passion in my work, love what I am doing “	2	7%
	academic success	“wellbeing means whether I can graduate or not”	2	7%
	self-value	“wellbeing relates to self-value”	2	7%

Note. Percentages add to more than 100, as responses could be classified into multiple categories.

**Table 3 ijerph-17-05538-t003:** Activities participants engage in to maintain their own wellbeing.

Categories	Major Themes	Sub Themes	Representative Quotes	*n*	%
intrapersonal activities 76%	cognitive activity 31%	learning & exploring	“enhance capability, “try new things”	48	11%
		study & work	“study and work hard”	34	8%
		planning & achieving	“stick with plans and achieve goals”	26	6%
		reflection	“change attitude”, “keep a diary”	25	6%
	physical activity 30%	exercise	“go to gym every week” “regular exercise”	71	16%
		healthy lifestyle	“get enough sleep”, “balanced diet”	45	10%
		meditation & relax	“plan to do meditation and yoga”	14	3%
		close to nature	“being in nature”. “enjoy the sunshine”	6	1%
	others 15%	hobbies	“watch movie”, “listen to music”	38	9%
		cooking & eating	“when I feel upset, I will eat something sweet”	14	3%
		self-protection	“try to avoid unsafe environment”	6	1%
		pet	“when I feel upset, I will stay with my cat”	3	1%
		self-harm	“drinking and stay up late”	1	<1%
interpersonal activities24%	relationship support 15%	friend	“talk to my friends”, “hang out with friends”	41	9%
		family	“talk to my parents on the phone”	22	5%
		partner	“spend time with my wife”	4	1%
	broad social support 7%	support others	“be a volunteer”, “help others with my ability”	6	2%
		spiritual support	“seek for religious help”, “go to church”	5	1%
		others	“chat with my teachers to improve wellbeing”	18	4%
	professional help 2%	counselling service	“go to counselling services”, “chat with counsellor”	6	1%
		campus security	“seek help from campus security”	1	<1%

**Table 4 ijerph-17-05538-t004:** Activities participant have or plan to engage in to support other’s wellbeing.

Categories	Main Themes	Subthemes	Representative Quotes	*n*	%
others-oriented 82%	activity-based 42%	do things together	“hang out”, “travel together”	36	15%
		support	“help others with what need”	31	13%
		company & care	“just hug my friends”, “company”	15	6%
		voluntary work	“be a volunteer to help others”	8	3%
		financial help	“donate to help poor women”	7	3%
	language-based 40%	speaking	“discuss their problems with them”	55	24%
		listening	“listening without interrupting”	37	16%
		writing	“write gratitude letters/messages”	1	<1%
self-oriented 18%	self-behaviours 17%	be better self & influence others	“do my best and influence others”	9	4%
		be kind & sensitive & empathy	“be kind and think in other’s shoes”	26	11%
		not trouble others	“don’t bring troubles to others”	4	2%
	nothing 2%	don’t care	“don’t care”, “nothing”	3	1%
		no further action if no change	“I will do nothing if no change”	1	<1%

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
