# Peer review of "Strengthening University Student Wellbeing: Language and Perceptions of Chinese International Students"

_ijerph, 2020, doi:10.3390/ijerph17155538_

Round 1

Reviewer 1 Report

The study is very interesting. More and more students are studying in third countries. Culture or language are big barriers for these students.

Especially students from oriental countries often have problems adapting to occidental countries.

The structure of the article is very clear. The appropriate methodology.

Focusing the study of analysis on well-being is very appropriate and we agree with its results and conclusions.

p.8: Do you think that the place of origin of China can be a variable to be analyzed?

p.9: Many foreign students when they go to a new country to study, they do not interact with the locals. Their relationships are only with other students from the same country.

Do you think that this can be a problem of insertion?

How do you think this problem can be avoided?

p.14: The questions have been asked in Chinese. Is this because of the low knowledge of the English language?

Don't you think you should identify language as one of the main problems for your integration?

p15: you said: “How to provide a safe and respectful environment for Chinese international students is an urgent issue for tertiary education institutions in Australia”. Do you really think this is a problem for Australian universities and they should resolve the problem? If so, tell us how to do it.

“The services around supporting students’ five component of wellbeing should be taken into consideration, such as offering healthy diet and lifestyle information, promoting mental health and counselling service, providing opportunity for cross-cultural interaction and activities, enhance campus security and non-discrimination environment, and educating students about employability opportunities”

If the main problem is the language and the second is the lack of interest of the foreign students in their integration, do you think that your recommendations would be enough?

The conclusions do not reflect all the work realized in its results. They should be expanded and the authors should clearly indicate each of the points analysed.

Author Response

Point 1: The study is very interesting. More and more students are studying in third countries. Culture or language are big barriers for these students. Especially students from oriental countries often have problems adapting to occidental countries. The structure of the article is very clear. The appropriate methodology. Focusing the study of analysis on well-being is very appropriate and we agree with its results and conclusions.

Response 1: We appreciate these positive comments, and believe that the results are timely and useful, especially considering growing concerns over mental health and wellbeing arising from the COVID-19 pandemic. We also appreciate the interesting questions and suggestions.

Point 2: p.8: Do you think that the place of origin of China can be a variable to be analysed?

Response 2: This is an interesting suggestion, but we have concerns around making assumptions based upon participants’ places of origin. Although different provinces have various social-economic conditions and local culture features, the individual’s perspective is more critical in this study. China has witnessed a mass migration in the past few decades, and lots of Chinese grown up in various provinces across north and south or west and east. Also, the widely used internet and social media promote mutual information and understanding between regions. Thus, it can be misguided to assume a person’s place of origin of China will represent or determine his/her own perspective on wellbeing, without directly asking participants, which we did not do.

Point 3: p.9: Many foreign students when they go to a new country to study, they do not interact with the locals. Their relationships are only with other students from the same country. Do you think that this can be a problem of insertion? How do you think this problem can be avoided?

Response 3: These are some fascinating questions, which unfortunately we cannot address through the data collected within our study, as they go beyond the intentions of the study. It is interesting to consider some of the insights that perhaps arise through our study. From the interview data, participants mentioned that they hope universities could provide more social opportunities to interact with locals and workshops to introduce local culture knowledge, such as slang, local foods, and festivals. They also mentioned that universities could provide more cross-culture activities to enhance the awareness of cultural diversity on campus, as some locals do not want to interact with international students when they been approached. We have added a paragraph in the discussion that speaks to this (see p. 15).

Point 4: p.14: The questions have been asked in Chinese. Is this because of the low knowledge of the English language? Don't you think you should identify language as one of the main problems for your integration?

Response 4: Participants had the option to choose to answer the questions in Chinese, English, or a mixture of languages. Most participants used a mixture of languages when completing the survey and interview. Some concepts appeared to be more easily discussed in Chinese, whereas other ideas appeared more accessible in English. Considering that all participants had an understanding of English at a high enough level that they could be admitted into top-level English-speaking universities, this suggest that they had a good understanding of the English language. We did not focus on integration; we focused on understandings of wellbeing. It is intriguing to consider why participants chose different languages at different times, and these will be important areas to explore in future studies.

Point 5: p15: you said: “How to provide a safe and respectful environment for Chinese international students is an urgent issue for tertiary education institutions in Australia”. Do you really think this is a problem for Australian universities and they should resolve the problem? If so, tell us how to do it.

Response 5: We do believe that creating safe and respectful environments are important, but this goes beyond the data arising from our study. Primary concerns over safety in our study arose from financial concerns, which most likely have been accentuated in the current context. Further, recent studies have pointed to concerns over discrimination that has occurred for Chinese people in Australia due to the Covid-19 pandemic. We have toned down our language, pointing to the issues here, but being cognizant that our study does not directly point to the urgency of this issue. We note in the discussion:

Aligned with the World Health Organization (1948), conceptualizations of health and wellbeing included physical, mental, and social aspects. For Chinese international students, a sense of security and prosperity also mattered. Unfortunately, a sense of security has been undermined for many Chinese internationals, due to discrimination and stigmatization that occurred through the COVID-19 pandemic [90]. This would suggest that it will be important for tertiary education institutions in Australia to identify the ways to provide safe and respectful environment for Chinese international students in the future.

Future research should consider how to resolve the problems that exist.

Point 6: The services around supporting students’ five component of wellbeing should be taken into consideration, such as offering healthy diet and lifestyle information, promoting mental health and counselling service, providing opportunity for cross-cultural interaction and activities, enhance campus security and non-discrimination environment, and educating students about employability opportunities” If the main problem is the language and the second is the lack of interest of the foreign students in their integration, do you think that your recommendations would be enough?

Response 6: These are important questions to be asking, which go beyond the data available in our study. Most likely, our recommendations are insufficient but at least a helpful start. We have adjusted our language to indicate that our suggestions are a starting point, within the limitations around representation noted, but which hopefully can spark additional research and guidance in terms of what recommendations are needed.

Point 7: The conclusions do not reflect all the work realized in its results. They should be expanded, and the authors should clearly indicate each of the points analysed.

Response 7: Our intention in the conclusion is to bring the paper to a close, not to provide a summary of all of the findings, which we provide at the beginning of the discussion, and expand upon throughout the discussion. As such, we’d prefer to keep the conclusion clear and concise, but are happy to modify further with recommendations around the extent to which findings should be re-summarised and expanded upon.

Reviewer 2 Report

The topic of the manuscript is interesting and important for university to support students study abroad. 84 Chinese international students completed and online survey and a sub-set of 30 students participated in semi-structured interviews. The manuscript studied reveal language that Chinese students use to talk about wellbeing and strategies they use to support wellbeing. This study broaden the understanding the wellbeing concept. However, there are two suggestions for the authors as follows.

1. Please explain the meaning of Figure 4. Why did authors demonstrate and compare the different understanding? Is the difference critical to understand the welling concepts of international students? How do universities provide supports based on the difference?
2. In section 4.3, the authors indicated tertiary education institutions should provide a respectful environment for Chinese students. I believed this is quite critical for foreign students. However, the authors didn’t provide reasons from data analysis results in Figure 1 and 2. Please explain why respectful environment is an urgent issue.

Author Response

Point 1: The topic of the manuscript is interesting and important for university to support students study abroad. 84 Chinese international students completed and online survey and a subset of 30 students participated in semi-structured interviews. The manuscript studied reveal language that Chinese students use to talk about wellbeing and strategies they use to support wellbeing. This study broadens the understanding the wellbeing concept.

Response 1: We appreciate the positive and supportive comments, and do believe that the study provides a helpful broader understanding of the wellbeing concept.

Point 2: However, there are two suggestions for the authors as follows.

Response 2: Thank you for the suggestions, as we believe the manuscript has been strengthened by comments by the reviewers.

Point 3: Please explain the meaning of Figure 4. Why did authors demonstrate and compare the different understanding? Is the difference critical to understand the welling concepts of international students? How do universities provide supports based on the difference?

Response 3: Participants could use English, Chinese, or a mixture, and we believe that it is important to consider what conceptualisations occur when they choose to use the different languages. The figure points to different ways that wellbeing is understood, based on the words used. We do believe that this is critical to consider, both theoretically and practically, as it broadens the understanding of wellbeing concept, providing a linguistic angle to consider the underpinning culture beliefs and values. For universities, Figure 4 might provide guidance in terms of how to message wellbeing-related information to students, and how that might differ depending on whether English or Chinese materials are provided, enabling support services to be more culturally sensitive. We have revised the description in the text to better clarify the information that we believe the figure provides. We also further clarify this addition in the discussion (see p. 13).

Point 4: In section 4.3, the authors indicated tertiary education institutions should provide a respectful environment for Chinese students. I believed this is quite critical for foreign students. However, the authors didn’t provide reasons from data analysis results in Figure 1 and 2. Please explain why respectful environment is an urgent issue.

Response 4: Reviewer 1 had a similar concern. We do believe that it is a critical issue considering research around the discrimination that has occurred during the pandemic, but calling this an urgent issue arose from our own beliefs, rather than from the data itself. We toned down our language. However, this is coming from recent research, which complements our findings, rather than coming from our research itself. We did tone down our language noting in the discussion, noting that the recent findings on discrimination that has occurred through the COVID-19 pandemic suggest that it will be important for tertiary education institutions in Australia to identify the ways to provide safe and respectful environment for Chinese international students.

Reviewer 3 Report

This is an interesting study on wellbeing literacy for Chinese international students from the students’ perspective. International students are perceived as a high-risk group for mental health and wellbeing issues because of transition issues. This study contributes to improvement in Chinese international student’s wellbeing through looking at how the concept of wellbeing is perceived from a bilingual perspective.

Editing

Abstract - 84 Chinese international students completed and online survey

p. 1 The coronavirus pandemic across 2019 and 2020 has brought public health into the spotlight, with tremendous recognition of how much physical health matters.

p. 2 public health and medicine have made excelled at preserving life, the quality of that life must also be considered [3].

p. 3 punctuation - that might have been useful in the home country, increases risk for a number of mental health issues, [39].

p. 4 Studying overseas is a critical period of young adults, which impacts health and development throughout the life course.

p. 5 Chinese International Students Associations and other this type of associations/student groups of the eight universities in Melbourne were contacted via email and WeChat, inviting them to share study details with eligible students

Caption for Figures 1 and 2 could be shortened and explanations placed in the text.

p. 14 This studied reveal language that Chinese international students use to understand and talk about wellbeing, as well as strategies they use to support wellbeing in themselves and others

p. 15 The wellbeing pathways revealed in this study offer message to tertiary institutions on what students actually may need to support their wellbeing, which might differ from the typical services that are offered.

Author Response

Point 1: This is an interesting study on wellbeing literacy for Chinese international students from the students’ perspective. International students are perceived as a high-risk group for mental health and wellbeing issues because of transition issues. This study contributes to improvement in Chinese international student’s wellbeing through looking at how the concept of wellbeing is perceived from a bilingual perspective.

Response 1: We appreciate the positive comments, which describe the focus and contributions well. We also appreciate the editing suggestions, which we have correct as noted below.

Point 2: Abstract - 84 Chinese international students completed and online survey

Response 2: Thank you, we have corrected this to say: “84 Chinese international students completed the online survey.”

Point 3: p. 1 The coronavirus pandemic across 2019 and 2020 has brought public health into the spotlight, with tremendous recognition of how much physical health matters.

Response 3: We are quite clear on the suggestion being made here, but have reworded this for greater clarity: “The coronavirus pandemic of 2019 and 2020 has brought public health into the spotlight, emphasising the extent to which physical health matters.”

Point 4: p. 2 public health and medicine have made excelled at preserving life, the quality of that life must also be considered [3].

Response 4: We have corrected this to note: “although public health and medicine have excelled at preserving life, the quality of that life must also be considered [3].

Point 5: p. 3 punctuation - that might have been useful in the home country, increases risk for a number of mental health issues, [39].

Response 5: We have corrected this to note: “that might have been useful in the home country, increases risk for a number of mental health issues [39].”

Point 6: p. 4 Studying overseas is a critical period of young adults, which impacts health and development throughout the life course.

Response 6: Corrected to note: “Studying overseas provides numerous opportunities for young adults, but also brings a number of challenges. There is a persistent gap…”

Point 7: p. 5 Chinese International Students Associations and other this type of associations/student groups of the eight universities in Melbourne were contacted via email and WeChat, inviting them to share study details with eligible students

Response 7: Corrected to note: “Chinese International Students Associations and other similar associations/student groups from the eight universities in Melbourne were contacted via email and WeChat, with an invitation to share the study details with eligible students”

Point 8: Caption for Figures 1 and 2 could be shortened and explanations placed in the text.

Response 8: We have shortened these captions, adding the explanation to the text and removing unnecessary words.

Point 9: p. 14 This studied reveal language that Chinese international students use to understand and talk about wellbeing, as well as strategies they use to support wellbeing in themselves and others

Response 9: We have corrected this to say: “This study reveals language that Chinese international students use to understand and talk about wellbeing, as well as strategies that they use to support wellbeing in themselves and others.”

Point 10: p. 15 The wellbeing pathways revealed in this study offer message to tertiary institutions on what students actually may need to support their wellbeing, which might differ from the typical services that are offered.

Response 10: Corrected to say: “The wellbeing pathways revealed in this study can inform tertiary institutions on what students actually may need to support their wellbeing, which might differ from the typical service that are offered.”

Round 2

Reviewer 2 Report

The author has made the necessary changes suggesting by the reviewer’s recommendations, and I think this article could be accepted.